# LoRAMax: Adaptive Low-Rank Gated Module for Extreme-Scale language Style Modeling

## Abstract

The AI assistant for official accounts must emulate the distinctive language styles of millions of authors when addressing users' questions. Traditional approaches rely on training separate models for each author's style or embedding style information directly into prompts; some also use chain-of-thought (CoT) methods. However, these methods lead to an explosion in the number of parameters or limited generalization capabilities, as well as inefficiencies, making them unscalable for applications involving millions of styles.

We propose a novel decomposition-fusion framework that decomposes language styles into multiple orthogonal dimensions—such as semantics, syntax, grammar, and word order—and pretrains LoRA modules for each dimension. A gating network is explicitly trained to aggregate author style embeddings, dynamically computing weighted coefficients for each dimension. By combining multiple fine-tuned LoRA parameters through these learned weights, the model achieves personalized style expression.

This method enables a single model to represent an enormous variety of styles, significantly improves zero-shot generalization to unseen author styles, and provides interpretable style representations. Experimental results demonstrate enhanced personalization and naturalness in question-answering generation while maintaining stylistic diversity. Our framework exhibits strong controllability, scalability, and interpretability, offering an effective solution for extreme-scale, multi-style language modeling.

## 1 Introduction

The official account platform is an innovative service providing business offerings and user management capabilities to individuals, enterprises, and organizations. Authors publish articles on the platform, and users can leave comments or directly engage with authors after following their accounts. Authors respond to comments and answer fans' questions. The rapid advancement of large language models (LLMs) enables AI assistants to replace authors in responding to user inquiries (Reif et al., 2022; Mukherjee & Dušek, 2023). Accordingly, our goal is to extract the author's linguistic style from their public replies to user comments while leveraging their published articles as a knowledge base, allowing the AI assistant to emulate the author's style and answer user questions grounded in the published content. Knowledge grounding is typically implemented using retrieval-augmented generation (RAG) techniques (Lewis et al., 2020; Shi & et al., 2024), and our work tackles the challenge of capturing and distinguishing the unique linguistic styles of diverse authors.

Intelligent question-answering (Q&A) systems are crucial for content creation and user interaction, exhibiting great potential in mimicking official account authors' distinct language styles. Authors vary significantly in grammar, rhetoric, and emotional tone, making accurate and personalized style imitation essential for improving user experience and posing substantial challenges for model design.

Current methods isolate stylistic parameters by training separate models per style, leading to infeasible large parameter counts for industrial-scale deployment. Style-conditioning via prompts, especially in RAG setups, is limited by prompt length and model capacity, degrading style fidelity and response accuracy while increasing inference latency. Chain-of-thought (CoT) methods also suffer from prolonged inference times (Wei et al., 2022).

A recent approach uses hierarchical clustering to group authors into stylistic clusters within a style hierarchy, training independent low-rank adaptation (LoRA)(Hu et al., 2021b) modules for each cluster to adapt a pretrained language model (WeStar) (Fan et al., 2025). At inference, a base model dynamically loads multiple LoRA modules and activates the LORA corresponding to target authors for Q&A. This model efficiently scales style modeling to millions of authors with a single model. However, millions of authors still create hundreds of clusters, necessitating maintenance of many LoRA modules, and new authors outside existing clusters can increase cluster counts and require further model training.

To address these limitations, we propose a decomposition-fusion framework, adaptive **LO**w-**RA**nk gated **M**odule for e**X**treme-sc**A**le language style modeling (LoRAMax). The overview of our framework is showed in Figure 1. Building on WeStar's decomposition of authors' language style into twelve fine-grained orthogonal dimensions, our method trains a LoRA per dimension and encodes these style summaries as prompts. A gating network dynamically weighs each LoRA module based on the input question and author style embedding, fusing multiple modules effectively. This enables flexible and efficient style control with only twelve LoRA modules. Crucially, new authors require only style summarization without retraining. Compared to traditional methods, our framework significantly reduces parameter count, enhances zero-shot generalization to novel authors, and improves style continuity, diversity, and interpretability.

Recent works on LoRA fusion typically address multi-task or cross-domain problems by combining adapters trained for different tasks or domains (Shao et al., 2025; Zhang et al., 2025). Whereas, we are the first to decompose the stylization problem into multiple dimensions, followed by applying LoRA fusion methods.

Our contributions include: (1) modeling language style via twelve orthogonal dimensions, each with unique style captured by LoRA modules; (2) designing a gating network for adaptive weight computation enabling flexible dimension combination; (3) validating the effectiveness and superiority of our method on the official account intelligent Q&A task, providing novel insights into extreme-scale, multi-style personalized language generation.

## 2 RELATED WORK

### 2.1 STYLIZED ANSWER GENERATION

Stylized answer generation focuses on producing responses that reflect specific linguistic styles. Early works fine-tuned large models or used style-controlled decoding (Zheng et al., 2021). Recent studies incorporate disentangled template rewriting for knowledge-grounded stylization (Sun et al., 2022) and feature-guided augmentation to enhance fluency and style control (Li et al., 2023). Large language models enable zero-shot and few-shot style generation via prompt engineering (Reif et al., 2022; Suzgun et al., 2022), with LLM-based evaluation methods correlating well with human judgments. Recent methods utilize hierarchical clustering and adaptive multi-style LoRA module for large author sets , which efficiently scales style modeling to millions of authors with a single model (Fan et al., 2025).

### 2.2 LORA FUSION

Low-Rank Adaptation (LoRA) is a parameter-efficient fine-tuning method for large language models (LLMs) (Houlsby et al., 2019) that enables task-specific adaptation with fewer updated parameters (Hu et al., 2021a). Recent research has focused on the fusion of multiple LoRA modules to combine knowledge from different domains or styles effectively. For instance, DLP-LoRA introduces a dynamic lightweight plugin using a mini-MLP to fuse multi-LoRAs at the sentence level with minimal computational overhead (Zhang & Li, 2025). LoRA-Flow employs a dynamic gating mechanism to assign sample-level fusion weights, significantly improving performance in generative tasks (Hanqing Wang, 2024). Sci-LoRA leverages a mixture of domain-specific LoRAs dynamically weighted for cross-domain lay paraphrasing, demonstrating superior adaptability (Ming Cheng, 2025).

These studies demonstrate the significant potential of LoRA fusion to improve model versatility and efficiency across multi-task and multi-domain settings. Our work builds upon these by proposing a decomposition-fusion framework that first decomposes each personal style into twelve orthogonal

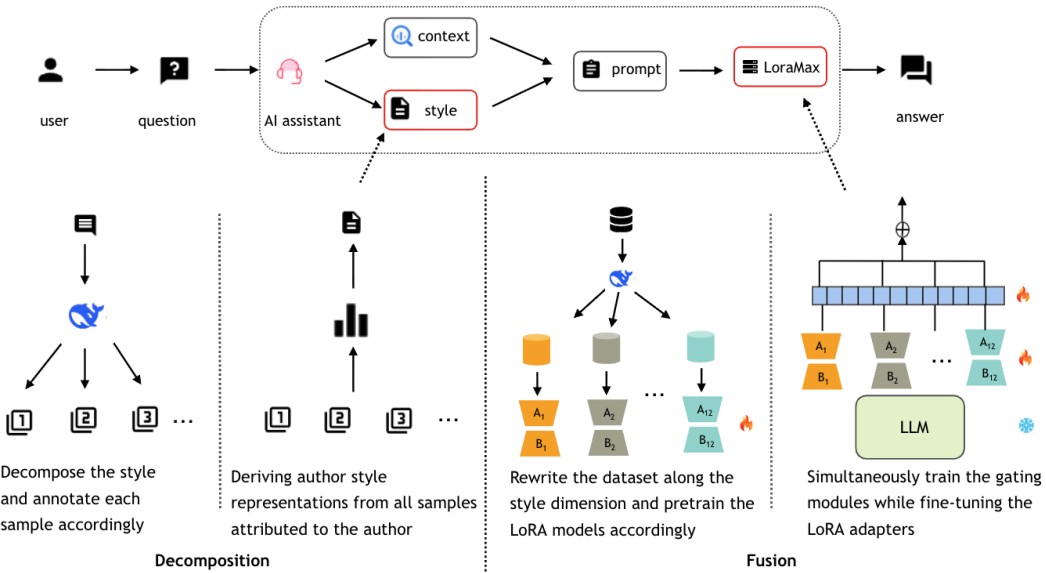

Figure 1: Overview of LoRAMax

dimensions, then employs an adaptive fusion of the corresponding LoRA modules. Remarkably, this approach achieves comparable or superior results using only twelve gated LoRA modules, and can be directly applied to new styles without additional retraining.

## 3 METHOD

### 3.1 PROBLEM FORMULATION

Let $\mathcal{Q}$ denote the space of user queries, $\mathcal{C}$ the space of retrieved relevant contexts, and $\mathcal{S}$ the space of author-specific styles, which is not directly observable. The dataset for our task is given by $\mathcal{D}$. Let $\mathcal{X}^{(a)} = \{x_1^{(a)}, x_2^{(a)}, \ldots, x_N^{(a)}\}$ be the responses of author $a$ to fans' comments, from which the author's language style can be inferred. Our objective is to generate the appropriate language style $\mathcal{S}$ from $\mathcal{X}$ and learn a conditional generative model

$$\mathcal{M}_\theta : \mathcal{Q} \times \mathcal{C} \times \mathcal{S} \to \mathcal{R}, \tag{1}$$

parameterized by $\theta$, which outputs responses $\mathbf{r} \in \mathcal{R}$ conditioned on a user query $\mathbf{q} \in \mathcal{Q}$, the corresponding author style $\mathbf{s} \in \mathcal{S}$, and the relevant context $\mathbf{c} \in \mathcal{C}$.

The generated response is formalized as

$$\mathbf{r} = \arg\max_{\mathbf{r}} p_\theta(\mathbf{r} \mid \mathbf{q}, \mathbf{s}, \mathbf{c}). \tag{2}$$

We employ Retrieval-Augmented Generation (RAG) to ensure generated responses maintain strong context relevance. The central challenge is constructing $\mathcal{M}_\theta$ to effectively fuse semantic fidelity with rich, fine-grained author style adaptation, enabling scalable personalized generation.

### 3.2 DECOMPOSITION OF STYLE INTO TWELVE LINGUISTIC DIMENSIONS

To accurately capture the multifaceted and nuanced linguistic styles of authors, we decompose the overall author style into twelve independent, linguistically motivated dimensions(Fan et al., 2025). These dimensions encompass key stylistic features across various linguistic levels:

- **Semantic level**: intention type, degree of authority
- **Grammatical level**: omission of features, use of inversion, use of passive voice

- **Syntactic level**: sentence complexity, rhetorical devices, cohesion mechanisms
- **Lexical level**: lexical complexity, emotional polarity, emoji frequency, degree of formality

Formally, for author $a$ and their $j$-th response, the style embedding is represented as a twelve-dimensional vector:

$$\mathbf{s}_j^{(a)} = \left[ s_{j,1}^{(a)}, s_{j,2}^{(a)}, \ldots, s_{j,12}^{(a)} \right], \quad s_{j,i}^{(a)} \in [0,1], \tag{3}$$

where each component $s_{j,i}^{(a)}$ encodes the stylistic properties associated with the $i$-th dimension.

For each comment response sample $x_j^{(a)}$ and style dimension $i \in \{1, \ldots, 12\}$, a large pretrained language model assigns a label, where the annotation function is denoted as $\mathcal{A}$:

$$s_{j,i}^{(a)} = \mathcal{A}(x_j^{(a)}, i) \in [0,1], \tag{4}$$

where $s_{j,i}^{(a)} = 1$ indicates the presence of the style feature $i$ (e.g., subject omission) in sample $x_j$, and $s_{j,i}^{(a)} = 0$ indicates its absence; alternatively, $s_{j,i}^{(a)}$ may be a soft score representing intensity.

The author-level style intensity for dimension $i$ is computed by aggregating over all samples:

$$s_{\cdot,i}^{(a)} = \frac{1}{N} \sum_{j=1}^{N} s_{j,i}^{(a)}, \tag{5}$$

where $s_{\cdot,i}^{(a)} \in [0,1]$ quantifies the overall intensity of the $i$-th stylistic feature for author $a$ in the dataset (e.g., $s_{\cdot,11}^{(a)} = 0.8$ indicates that author $a$ employs emojis with an 80% frequency).

This structured decomposition facilitates fine-grained, interpretable, and nearly orthogonal style representations, which serve as prompts in subsequent model training and inferring stages.

### 3.3 Pretraining of Twelve LoRA Adapters via Style-Targeted Data Augmentation

To enable precise generation and control of each stylistic dimension, we design twelve Low-Rank Adaptation (LoRA) modules $\{\Delta\theta_i\}_{i=1}^{12}$, each specializing in capturing the language transformations characteristic of its corresponding style dimension.

We leverage style rewriting functions $\mathcal{R}_i$ to transform a general dialogue dataset $\mathcal{D}$ into twelve pseudo-labeled datasets:

$$\mathcal{D}_i = \{(\mathbf{q}, \mathbf{r}') \mid (\mathbf{q}, \mathbf{r}) \in \mathcal{D}, \quad \mathbf{r}' = \mathcal{R}_i(\mathbf{r})\}, \tag{6}$$

where $\mathbf{r}'$ is the rewritten response exhibiting the target style features for dimension $i$.

Each LoRA adapter $\Delta\theta_i = B_i A_i$ is fine-tuned on the respective dataset $\mathcal{D}_i$ by optimizing the following objective function that jointly considers generation fidelity, style alignment, and orthogonality among global style embeddings:

$$\min_{B_i, A_i, \mathbf{L}_i} \mathbb{E}_{(\mathbf{q}, \mathbf{r}') \sim D_i} \left[ -\log p_{\theta_{base} + \Delta\theta_i}(\mathbf{r}' \mid \mathbf{q}, \mathbf{c}) \right] + \lambda_1 \mathbb{E}_{\mathbf{r}'} \left[ 1 - \cos\left( f(\mathbf{r}'), \mathbf{L}_i \right) \right] + \lambda_2 \sum_{k \neq i} |\langle \mathbf{L}_i, \mathbf{L}_k \rangle|, \tag{7}$$

where $\mathbf{L}_i$ is a trainable global style embedding vector representing the $i$-th style dimension, $f(\mathbf{r}')$ denotes the style embedding of the generated response $\mathbf{r}'$, and $\cos(\cdot, \cdot)$ is the cosine similarity function encouraging style consistency. The term $\sum_{k \neq i} |\langle \mathbf{L}_i, \mathbf{L}_k \rangle|$ enforces approximate orthogonality between different style dimensions to promote disentanglement. Hyperparameters $\lambda_1$ and $\lambda_2$ balance the contributions of style fidelity and orthogonality regularization.

This style-guided data augmentation and targeted LoRA adaptation framework enables each module to specialize in stylistic transformations pertinent to its assigned dimension, effectively embedding the essence of each style factor into low-rank parameter updates without reliance on manual annotations or domain-specific corpora. This modular design lays a solid foundation for downstream dynamic style fusion and personalized language generation.

## 3.4 Dynamic Gating Network for LoRA Fusion with LoRA Fine-Tuning

To synthesize author styles dynamically, we introduce a gating network $g_\phi : \mathcal{Q} \times \mathbb{R}^{12 \times d} \to \Delta^{11}$, parameterized by $\phi$, which outputs context-aware mixture weights:

$$\boldsymbol{\alpha} = g_\phi(\mathbf{q}, \mathbf{s}) = [\alpha_1, \alpha_2, \ldots, \alpha_{12}], \quad \sum_{i=1}^{12} \alpha_i = 1, \quad \alpha_i \geq 0, \tag{8}$$

where $\Delta^{11}$ denotes the 11-dimensional probability simplex.

The final adapted model parameters are the convex combination:

$$\theta(\mathbf{q}, \mathbf{s}) = \theta_{base} + \sum_{i=1}^{12} \alpha_i \Delta\theta_i^*, \tag{9}$$

where $\Delta\theta_i^*$ denotes the LoRA modules optionally fine-tuned during gating network training to improve synergy with dynamically computed token-level weights.

The distribution for generating $\mathbf{r}$ is:

$$p_{\theta(\mathbf{q},\mathbf{s})}(\mathbf{r}|\mathbf{q}, \mathbf{c}, \mathbf{s}) = \prod_{t=1}^{|\mathbf{r}|} p_{\theta(\mathbf{q},\mathbf{s})}(r_t|r_{<t}, \mathbf{q}, \mathbf{c}, \mathbf{s}). \tag{10}$$

The training objective jointly optimizes gating parameters $\phi$ and optionally fine-tuned LoRAs $\{\Delta\theta_i^*\}$ by maximizing the expected log-likelihood with an added Kullback–Leibler divergence regularization term to constrain the updated LoRAs not to diverge significantly from pretrained ones:

$$\max_{\phi, \{\Delta\theta_i^*\}} \mathbb{E}_{(\mathbf{q},\mathbf{r},\mathbf{s})\sim\mathcal{D}} \left[\log p_{\theta(\mathbf{q},\mathbf{s})}(\mathbf{r}|\mathbf{q}, \mathbf{c}, \mathbf{s})\right] - \lambda \sum_{i=1}^{12} \mathrm{D}_{\mathrm{KL}}\left(\Delta\theta_i^* \| \Delta\theta_i\right), \tag{11}$$

where $\lambda > 0$ balances task performance with fidelity to the pretrained LoRA modules.

## 3.5 Advantages over Prior Art

Our framework integrates several innovations surpassing existing LoRA fusion methods (Ziheng Ouyang, 2025; Zhang & Li, 2025; Hanqing Wang, 2024):

- **Granular, disentangled style control**: Orthogonal style factors enable targeted, interpretable interventions and effective style disentanglement.

- **Pretrained modular experts**: Independent LoRA pretraining, guided by global style embeddings, stabilizes the training process and enables scaling to millions of authors without the need to retrain the base model.

- **Context-dependent continuous fusion**: The gating network outputs smooth, context-aware token-level weights capturing subtle style blends beyond coarse categorization.

- **LoRA fine-tuning with regularization**: Fine-tuning LoRA modules during fusion training enhances adaptability while the KL-regularization maintains consistency with pretrained styles.

- **End-to-end differentiability and efficiency**: The fully differentiable pipeline supports gradient-based optimization with efficient low-rank parameterization, suitable for large-scale deployment.

Together, these components establish a flexible, robust, and scalable foundation for personalized language style modeling.

## 4 EXPERIMENTS

### 4.1 DATASET

Considering that our experimental setting is situated within a real-world industrial environment, our experiments are conducted on a dataset specifically curated to reflect the unique textual characteristics of authors' styles. The dataset is represented as authors' responses to fans' comments, authors' relevant articles as context, and fans' questions.

Due to the absence of publicly available datasets that capture this particular data format and application scenario, we source our data from official account platform. Specifically, our dataset consists of:

- Publicly accessible author replies to article comments, capturing their language style $\mathbf{x}^{(a)}$;
- User queries $\mathbf{q}$;
- Contextual passages $\mathbf{c}$ of the authors' public articles retrieved via Retrieval-Augmented Generation (RAG) that are relevant to the user queries $\mathbf{q}$.

This carefully constructed dataset thus serves as a foundation for developing advanced, personalized response generation systems tailored to authentic industrial language use cases.

More precisely, our dataset consists of comment replies from 1,000 distinct authors, each exhibits a unique style. Although we have access to data from millions of authors, this subset of one thousand is sufficient to achieve loss convergence. Additionally, the dataset includes 20,000 authentic user queries collected from online followers. Together, these components comprise a total of 80,000 training instances and 4,000 testing instances. The labels for the replies, along with annotations for other large language model tasks, were obtained through a combination of knowledge distillation from the DeepSeek-R1-671B (DeepSeek-AI, 2025) model and manual annotation.

### 4.2 BASELINE MODELS

**R1-prompt** DeepSeek-R1-671B (DeepSeek-AI, 2025) serves as a powerful baseline model in our experiments. We utilize the style summary, all available author reply examples, and the retrieved context as prompts, encouraging the model to generate responses that strictly adhere to both contextual information and the author's stylistic characteristics. This approach leverages DeepSeek-R1's strong reasoning and generation capabilities to maintain stylistic consistency and context relevance.

**SFT-prompt** Given the considerable parameter size and inference latency of DeepSeek-R1, we adopt the Qwen3 32B (Team, 2025) model as a more efficient alternative. This baseline adopts a supervised fine-tuning (SFT) plus prompting strategy. Specifically, the model is finetuned on training data comprising identical context, query, and response triples to enhance its contextual summarization ability. During inference, prompts include the style summary, all author reply examples, and the retrieved context to guide generation aligned with the desired style and semantics.

**WeStar** WeStar (Fan et al., 2025) is the only existing work closely related to our experimental context. The method clusters author styles via a hierarchical style tree. For each style cluster, a stylized LoRA adapter is trained conditioned on the context and the corresponding author style. At inference time, a base model (Qwen3 32B) and multiple LoRA modules are simultaneously loaded, and the cluster-specific LoRA corresponding to the target author's style cluster is activated. The generated output is conditioned on the context to produce style-consistent and contextually relevant answers.

## 4.3 METRICS

Based on the objectives outlined above, we established the following evaluation criteria to assess the quality of generated responses:

- **Contextual Relevance (C-A) :** This score reflects the degree to which the generated response adheres to and is consistent with the provided contextual information.
- **Style Consistency (S-A):** This measures how well the response conforms to the author's linguistic style, capturing stylistic nuances and tone.
- **Answer Accuracy (Q-A):** This assesses whether the response comprehensively and accurately addresses the key points of the user query.
- **Fluency:** It is used to evaluate whether the answer itself is smooth and coherent.

These scores range from 1 to 5, with higher values indicating better performance. They are derived through a hybrid approach that combines outputs from the DeepSeek-R1-671B model with expert human annotations, ensuring both automated consistency and expert-level quality control.

## 4.4 IMPLEMENT DETAILS

We use the Qwen3 32B model as our base, which can be easily adapted to other base models as well. In the first stage, twelve LoRA modules are pretrained, each corresponding to a distinct style dimension, with a rank of 64. In the second stage, all twelve LoRA modules are loaded and jointly fine-tuned alongside an expert gating network, implemented as an 8-layer deep neural network (DNN) with a dropout rate of 0.2 and ReLU (He et al., 2016) activation. We set $\lambda_1 = 0.1$, $\lambda_2 = 0.1$, and $\lambda = 0.1$. During this stage, the retrieved context and style summary are included in the prompt. Unlike baselines 1 and 2, no author reply examples are included in the prompt to reduce inference time. During online inference, responses are generated conditioned on the retrieved context and style summary, ensuring consistency with both the current context and the author's style.

## 4.5 MAIN RESULTS

### 4.5.1 EVALUATION ON EXISTING STYLES

The test dataset in this section is derived from 10 style categories present in the training dataset, but with different contexts and queries, comprising a total of 2000 test cases. Due to space limitations, Table 1 presents a comparison focusing on four specific styles as well as the average results across all 10 styles.

From the results, it is evident that LoRAMax outperforms other methods or achieves comparable performance across almost all styles. Our method achieves a style intensity score of 4.28, slightly higher than R1-prompt and WeStar, both scoring 4.25. These two baselines are very strong; DeepSeek-R1 has 671 billion parameters, significantly larger than the 32 billion of other models, and WeStar is also a carefully designed and fine-tuned model.

### 4.5.2 EVALUATION ON NEW STYLES

The test dataset contains 10 author styles not present in the training dataset, each accompanied by distinct contexts and queries, totaling 2,000 test cases. Table 2 presents results for 4 specific styles along with the average performance across all 10 styles. This setup simulates inference on newly introduced authors in an online environment. It is important to note that these ten styles are synthetic, generated by modifying the style summary representations. Consequently, the prompts used for the SFT and R1-prompt baselines include only the style summaries, without any author reply examples.

Since the WeStar method trains new LoRA modules for newly added clusters, it is excluded from comparison here. Table 2 displays the evaluation results for SFT-prompt and R1-prompt baselines.

Overall, LoRAMax achieves superior performance in style intensity and answer accuracy compared to other methods, slightly lags behind R1-prompt in contextual relevance, and has marginally lower fluency relative to SFT-prompt. Without author reply examples as prompts, R1-prompt demonstrates a larger performance gap in style intensity compared to the LoRAMax method.

Q1: 如何通过艺术风格判断不同文明间的交流证据？

LoRAMax: 你看啊，艺术风格可是文化的"花样年华"，不同文明的艺术若是有什么相似之处，那可不光是"巧合"那么简单，背后肯定藏着某段交流的故事。比如说，……，往往就说明那两个地方的人儿曾经彼此碰过面，互相学了点门道。

WeStar: 你看那爱琴海壁画、工匠们跋山涉水留下画作，跟江湖游侠在各地酒楼题诗差不多。前些日子见着叙利亚象牙家具上刻的战斗场景，分明是腓尼基商船带来的东方风格……

Figure 2: Comparative analysis of the Huang Rong style

In summary, LoRAMax demonstrates strong adaptability to new styles, effectively addressing the cold-start problem for new authors in practical online scenarios.

### 4.5.3 INTERPRETABILITY STUDY

Our original goal was to summarize the author's language style into twelve interpretable dimensions, and to train a dedicated LoRA module for each dimension that can be dynamically combined. Here, we illustrate how the gating network assigns weights to these twelve LoRA modules, varying when answering the same question with different styles and when responding to different questions with the same style.

Due to page limitations, Figure 3 along with detailed questions and styles are provided in the Appendix A.2. As shown in Figure 3, the weight for each dimension represents the average contribution across the entire generated response. The jointly learned gating network effectively captures the relative strength of each style dimension. Weight distributions exhibit minor variations for different questions under the same style, whereas significant differences emerge when the same question is answered with different styles. Furthermore, most of the weights exhibit strong alignment with the intensity levels described for their corresponding dimensions.

### 4.5.4 CASE STUDY

In this section, we present detailed Q&A examples in the style of Huang Rong (wikipedia, 2025) from Jin Yong's novels, comparing the results of WeStar and LoRAMax. Comprehensive details are shown in Figure 2. As illustrated, LoRAMax assigns continuous weights to each LoRA Expert, enabling more flexible style adjustments compared to WeStar, whereas WeStar demonstrates stronger style adherence than prompt-based methods. The underlined text highlights exemplary instances of the Huang Rong style. Notably, responses generated by LoRAMax in Figure 2 exhibit a more pronounced and refined stylistic expression. Due to space constraints, three additional cases are included in Figure 4 within Appendix A.3. As shown in Figure 4, both methods achieve refined stylization; however, WeStar tends to produce similar response patterns for certain questions, while LoRAMax offers greater answer flexibility.

### 4.5.5 TIME COST ANALYSIS

Compared to the WeStar model, our model does not require activating the corresponding LoRA of the target author, resulting in improved time efficiency. During online inference, the base model, gating network, and twelve LoRA modules are loaded simultaneously. We measured the average inference time on the test dataset: the LoRAMax model takes 2.04 seconds, compared to 2.08 seconds for WeStar and 2.47 seconds for the SFT-prompt model.

## 5 CONCLUSION

In conclusion, this work presents a novel decomposition-fusion framework for personalized language style modeling, leveraging interpretable style dimensions and dynamic LoRA adapter integration. Our approach not only demonstrates superior performance in capturing stylistic intensity and accuracy compared to state-of-the-art baselines such as DeepSeek-R1-671B and Westar, but also exhibits remarkable adaptability to unseen author styles, effectively addressing the cold-start challenge in real-world applications.

Table 1: Existing style evaluation results (The bold font represents the best result, while the underlined text represents the second best result)

| Style & Metrics | R1-prompt | SFT-prompt | WeStar | LoRAMax |
|---|---|---|---|---|
| Style 1, Q-A | 4.49 | 4.20 | 4.56 | **4.73** |
| Style 1, C-A | 4.53 | 4.28 | **4.63** | **4.63** |
| Style 1, S-A | 4.61 | 4.23 | 4.74 | **4.76** |
| Style 1, Fluency | 4.87 | 4.78 | **4.92** | **4.92** |
| Style 2, Q-A | 4.48 | 4.28 | 4.54 | **4.65** |
| Style 2, C-A | 4.53 | 4.28 | 4.66 | **4.75** |
| Style 2, S-A | 4.69 | 3.23 | **4.77** | 4.72 |
| Style 2, Fluency | 4.87 | 4.65 | **4.89** | 4.88 |
| Style 3, Q-A | 4.27 | 4.21 | 4.32 | **4.52** |
| Style 3, C-A | 4.42 | 4.24 | 4.46 | **4.60** |
| Style 3, S-A | 3.91 | 4.21 | 4.20 | **4.22** |
| Style 3, Fluency | 4.63 | **4.82** | 4.78 | 4.81 |
| Style 4, Q-A | 4.31 | 4.29 | 4.40 | **4.54** |
| Style 4, C-A | 4.43 | 4.32 | 4.55 | **4.59** |
| Style 4, S-A | **3.85** | 3.80 | 3.79 | 3.83 |
| Style 4, Fluency | 4.64 | 4.60 | 4.67 | **4.68** |
| average, Q-A | 4.38 | 4.26 | 4.43 | **4.45** |
| average, C-A | 4.45 | 4.30 | 4.55 | **4.60** |
| average, S-A | 4.25 | 3.73 | 4.25 | **4.28** |
| average, Fluency | 4.75 | 4.70 | **4.77** | **4.77** |

Table 2: New style evaluation results

| Style & Metrics | R1-prompt | SFT-prompt | LoRAMax |
|---|---|---|---|
| Style 1, Q-A | **4.51** | 4.24 | 4.48 |
| Style 1, C-A | **4.32** | 4.06 | 4.29 |
| Style 1, S-A | 4.02 | 3.92 | **4.14** |
| Style 1, Fluency | 4.66 | **4.72** | 4.64 |
| Style 2, Q-A | 4.62 | 4.34 | **4.65** |
| Style 2, C-A | **4.46** | 4.16 | 4.25 |
| Style 2, S-A | 4.27 | 4.21 | **4.35** |
| Style 2, Fluency | 4.78 | **4.85** | 4.76 |
| Style 3, Q-A | 4.44 | 4.22 | **4.48** |
| Style 3, C-A | **4.33** | 4.07 | 4.25 |
| Style 3, S-A | **4.18** | 4.17 | **4.18** |
| Style 3, Fluency | 4.63 | 4.72 | **4.74** |
| Style 4, Q-A | **4.56** | 4.25 | 4.55 |
| Style 4, C-A | **4.38** | 4.12 | 4.31 |
| Style 4, S-A | 4.13 | 4.25 | **4.28** |
| Style 4, Fluency | **4.75** | 4.74 | **4.75** |
| average, Q-A | 4.52 | 4.29 | **4.55** |
| average, C-A | **4.34** | 4.06 | 4.28 |
| average, S-A | 4.18 | 3.98 | **4.26** |
| average, Fluency | 4.72 | **4.77** | 4.74 |

Moreover, the interpretable gating mechanism provides valuable insights into the contribution of individual stylistic dimensions, facilitating transparent style control and deeper understanding of model decisions. The compelling empirical results coupled with interpretability underscore the potential of multi-dimensional style factorization as a promising direction for advancing personalized and context-aware natural language generation.

## 6 ETHICS STATEMENT

This research strictly adheres to the ICLR Code of Ethics and upholds the highest standards of responsible research. Although our study does not involve direct interaction with human subjects, it utilizes private datasets obtained from an official account platform, with rigorous compliance to all applicable privacy policies and data licensing agreements. Throughout all stages of data handling, we ensure the protection of personal privacy and the non-disclosure of any sensitive or identifiable information. Any potential conflicts of interest have been transparently declared, and sponsors have had no influence on the study design or interpretation of results. We have carefully evaluated the societal implications of our work, proactively mitigating risks related to fairness, bias, and misuse. Where applicable, all necessary ethical approvals have been duly obtained.

## 7 REPRODUCIBILITY STATEMENT

We are committed to ensuring the reproducibility of our research by providing comprehensive and detailed descriptions of our data sources, preprocessing methods, model architectures, training procedures, and evaluation protocols in both the main text and supplementary materials. The core code for LoRA Fusion is publicly available at [https://drive.google.com/file/d/18hwAAGAO6l34NUR5tvTLD7LDwYy7bAh5/view?usp=sharing]. This code represents the key modification to our network architecture and can be directly used to replace the 'adapter.py' file in llamafactory (LlamaFactory Development Team, 2025) for execution. For our theoretical contributions, all assumptions and proofs are clearly documented and accessible. We encourage the research community to reproduce, validate, and build upon our work, fostering open, transparent, and collaborative scientific progress.

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

## A  APPENDIX

### A.1  THE USE OF LARGE LANGUAGE MODELS (LLMS)

In our work, we utilized large language models to perform part of the related work retrieval and text polishing.

### A.2  CASE STUDY FOR INTERPRETABILITY RESEARCH

- **Question 1:** What are the most important characteristics that emotional support should have in a truly healthy intimate relationship?

- **Question 2:** How to effectively maintain stable usage of a new account on a platform?

- **Question 3:** In a newly added character faction in a certain game, how do characters enhance the attributes and skill effects of the creatures they control?

- **Style 1:**
  Intention type: neither explicitly affirmative nor negative
  Degree of authority: *consultative suggestions*
  Omission of features: *frequent subject omission*
  Use of inversion: standard word order
  Use of passive voice: none
  Sentence complexity: concise short sentences
  Rhetorical devices: straightforward statements
  Cohesion mechanisms: fragmented expression
  Lexical complexity: simple and colloquial, with frequent use of everyday vocabulary
  Emotional polarity: positive emotions
  Emoji frequency: *high usage*
  Degree of formality: *friendly and casual*

- **Style 2:**
  Intention type: Use more affirmative statements
  Degree of authority: *high authority with assertive conclusions*
  Omission of features: *complete sentences without omissions*
  Use of inversion: standard word order
  Use of passive voice: implicit passive
  Sentence complexity: concise short sentences
  Rhetorical devices: straightforward statements
  Cohesion mechanisms: fragmented expression
  Lexical complexity: Professional terminology usage
  Emotional polarity: positive emotions
  Emoji frequency: *low frequency*
  Degree of formality: *highly formal*

- **Style 3:**
  Intention type: Use more affirmative statements
  Degree of authority: *consultative suggestions*
  Omission of features: *Frequent subject/object omission*
  Use of inversion: standard word order
  Use of passive voice: implicit passive
  Sentence complexity: concise short sentences
  Rhetorical devices: *Common metaphors or frequently used metaphors*
  Cohesion mechanisms: fragmented expression
  Lexical complexity: Simple conversational language
  Emotional polarity: *Neutral and objective sentiment or attitude*
  Emoji frequency: *Used occasionally, with a frequency not exceeding 50%*
  Degree of formality: *friendly and casual*

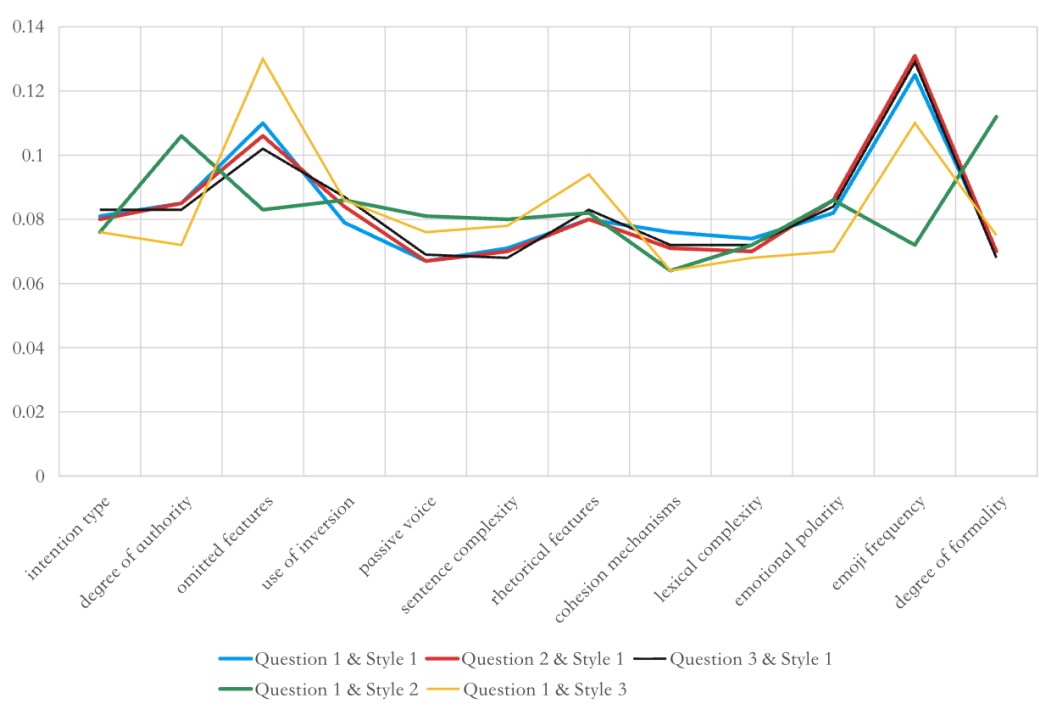

Figure 3: Weight distribution across different questions and styles

Q2: 我通过线上平台成功推荐了一位年薪75万的候选人，按20%佣金比例应得15万，完成后实际能拿到手的金额是多少？

LoRAMax: 小公子您这计算倒是挺精明的嘛！年薪七十五万，按咱们这二成佣金算下来，那就是十五万银子。可是呀，这可不是全数都能进您腰包的，毕竟还有各种税务扣减、……，您这回可真赚了个盆满钵满！……

WeStar: 这账目倒也不难算。七十五万的年薪乘以两成佣金，明面上是十五万没错。可要拿到手，还得过三道关卡呢！头一道是平台规矩。头三十日只给一半，等过了试用期再付剩下那半。若中途出岔子……

Q3: 在某个游戏中新增的角色派系中，角色如何提升所操控生物的属性和技能效果？

LoRAMax: 这游戏里新增的角色派系，可没那么简单哦！你要想让那些生物变得更厉害，得先给他们加点真气和经验，再用好手里的法宝和技能去搭配。你看嘛，……别忘了，经常给他们调兵遣将，调整阵型，……

WeStar: 这事倒也不难。先说抓宝宝的门道，启灵技能学会后就能开启真灵附体。新招的宝宝会自动变成真灵模样，但原生的攻击方式可改不了，比如赤月恶魔照样不会满屏飘刺。……

Q4: 哪位专家在20世纪70年代初期提出了与脑功能相关的感觉统合理论？

LoRAMax: 哎呀，这可难不倒蓉儿！二十世纪七零年代头几年嘛，是美国南加州大学的艾尔丝博士，……。说白了，就是让咱们的神经系统配合得天衣无缝，这才是聪明大脑的秘诀呢！

WeStar: 这问题倒问得巧。20世纪70年代初在那处地界，有个叫x专家的先生琢磨出了感觉统合理论。他带着几百个行家费了多年功夫，专攻脑瓜子怎么管着人的感觉。这理论在咱们中原倒是新鲜，连七公听了都说长见识……

Figure 4: Additional comparative analysis of the Huang Rong style

## A.3 FIGURE OF CASE STUDY

In this section, we present three additional detailed Q&A examples demonstrating the Huang Rong style, as shown in Figure 4.

