# OpenReview forum: "LoRAMax: Adaptive Low-Rank Gated Module  for Extreme-Scale language Style Modeling"
_ICLR.cc/2026/Conference — ICLR 2026 Conference Desk Rejected Submission_

### Official Review · Reviewer_p58k · 2025-10-29

**Soundness:** 2
**Presentation:** 2
**Contribution:** 2
**Rating:** 4
**Confidence:** 4

**Summary:**

This paper addresses the challenge of creating a single AI assistant capable of emulating millions of unique author styles for personalized question-answering. The authors propose LoRAMax, a novel decomposition-fusion framework. The core idea is to first decompose any author's style into a 12-dimensional vector representing predefined linguistic features. The framework then uses a fixed set of 12 "expert" LoRA modules, where each module is pretrained to specialize in manifesting one of these 12 style dimensions. At inference time, a gating network takes an author's 12-dimensional style embedding and the user query as input. It then computes a weighted sum of the 12 expert LoRA modules to create a final, style-specific adapter for the base LLM. Experiments on a large, private dataset show that LoRAMax can effectively model existing styles and, crucially, generalize to new, unseen author styles in a zero-shot manner, outperforming baseline methods like prompt-based style control and cluster-based LoRA models (WeStar).

**Strengths:**

1. Scalability and Efficiency: The primary strength of this work is its solution to the "extreme-scale" problem. Instead of training and storing millions (or even hundreds) of LoRA adapters (one per author or cluster), this framework only requires a fixed set of 12 adapters. This is a significant practical advantage for real-world deployment.

2. Zero-Shot Generalization: The framework's ability to handle new authors without any retraining is a strong contribution. A new author only needs their 12-dimensional style vector to be computed, which is a simple inference task, making the system highly adaptable.

3. Interpretability: The gating mechanism provides a clear, interpretable model. The weights assigned to each of the 12 LoRA modules (as shown in Figure 3) offer direct insight into which stylistic dimensions are being activated to construct the final response, which is valuable for analysis and debugging.

**Weaknesses:**

1. Lack of Novelty in Architecture: The core architectural concept of using a gating network to dynamically combine a set of expert LoRA modules is not new. This method is highly similar to existing work on Mixture of LoRA Experts (MoLE, Wu et al. 2024). The paper's contribution appears to be the application of this architecture to a set of pre-defined style dimensions, rather than a novel method in itself.

2. Incomprehensive Literature Review: The related work section is narrowly focused on LoRA fusion. It overlooks other relevant and recent lines of research in style control. For instance, methods using activation steering (e.g., Ma et al. 2025, "DRESSing up LLMs") also achieve fine-grained style adaptation and should be discussed and compared to properly position the paper's contribution. (e.g., 12 steering vectors for each of the linguistic features)

3. Unjustified Style Dimensions: The entire framework is built on the decomposition of style into 12 specific linguistic features. However, the paper states these dimensions are adopted from previous work (Fan et al. 2025). There is no justification provided for why these 12 features are chosen, whether they are comprehensive enough to capture the full spectrum of style, or if they are universally applicable.

4. Language and Dataset Limitations: The experiments are conducted exclusively on a private, Chinese-language dataset. This significantly limits the generalizability of the findings. Many of the 12 style features (e.g., "omission of features," "use of inversion") are highly language-dependent. The paper fails to discuss the linguistic-specific nature of these features in Chinese or demonstrate the framework's viability in other languages (like English). Moreover, authors should consider that non-Chinese-speaking readers are not able to understand the case, and provide an experiment in a more generally used language, such as English.

5. Vagueness in Pretraining: A critical component of the method is the pretraining of the 12 expert LoRA modules (Section 3.3). This step relies on "style rewriting functions ($\mathcal{R}_{i}$)" to create augmented data, but the implementation of these functions is not described. This lack of detail is a significant gap that hinders reproducibility.

6. Unsupported "Orthogonality" Claim: The paper claims the 12 style dimensions are "independent" and "nearly orthogonal." This is a strong claim that appears to be based only on an orthogonality-promoting regularization term in the loss function (Eq. 7), rather than any empirical validation. No analysis is provided to show how (in)dependent these learned style representations actually are.

**Questions:**

See Weaknesses

---

### Official Review · Reviewer_kw8c · 2025-10-31

**Soundness:** 1
**Presentation:** 2
**Contribution:** 2
**Rating:** 2
**Confidence:** 4

**Summary:**

The paper introduces a scalable and interpretable framework for generating text using personalized writing styles of millions of authors. It decomposes language style into twelve orthogonal linguistic dimensions (e.g., syntax, emotion, formality), trains one LoRA adapter per dimension, and fuses them dynamically through a gating network that computes context-dependent weights. This enables flexible, fine-grained style control and zero-shot generalization to unseen authors using only a small number of adapters. Experiments on an industrial, private Q&A dataset show that LoRAMax outperforms baselines (WeStar, DeepSeek-R1, and SFT-prompt) in style consistency, contextual relevance, and scalability, while maintaining interpretability and inference efficiency.

**Strengths:**

- Novel framework to do personalized style rewriting in a scalable manner.
- Empirical performance looks promising (but evaluation is questionable)

**Weaknesses:**

- Some things need clarification
	What is the “official account platform” (line 34)? Chinese examples not understandable for non-Chinese speakers
- Limited reproducibility: The experiments depend solely on a private dataset, which restricts both reproducibility and external validation. The source of the dataset is unclear
- Style and relevance scores partially depend on LLM-based scoring (DeepSeek-R1), which might introduce bias toward similar model families.
- There are no ablation studies that isolate the effects of the different parts.
- Questionable linguistic dimensions. The authors only cite the paper by Fan et al. 2025 (not even peer-reviewed) to motivate the chosen linguistic dimensions. NLP papers investigating linguistic style features have a long history, e.g.: McDonald, D. D., & Pustejovsky, J. (1985, March). A computational theory of prose style for natural language generation. In Second Conference of the European Chapter of the Association for Computational Linguistics.
- Questionable evaluation: Lines 337-339 describe how the authors mix human annotations and LLM annotations to get some final score. How does this hybrid approach exactly work?

**Questions:**

See limitations above:
- What is the “official account platform” (line 34)?
- Where does the dataset come from?
- How does the hybrid evaluation approach work exactly?

**Details Of Ethics Concerns:**

The first sentence of the paper is "The AI assistant for official accounts must emulate the distinctive language styles
of millions of authors when addressing users’ questions."
It sounds as if the authors have to manage some kind of AI assistant for an authoritarian regime where all user data is used for training style-specific AI assistants.

It is highly questionable whether it is ok to use the personal data of these unnamed users (data source is not disclosed).

---

### Official Review · Reviewer_cUZA · 2025-11-01

**Soundness:** 2
**Presentation:** 3
**Contribution:** 2
**Rating:** 2
**Confidence:** 4

**Summary:**

This paper investigates how to achieve scalable and interpretable personalization in large language models. The authors propose LoRAMax, a decomposition–fusion framework that represents language style through twelve orthogonal linguistic dimensions (e.g., semantics, syntax, emotion). Each dimension is captured by a specialized LoRA adapter, while a gating network dynamically fuses these adapters based on the query and author embeddings. Experimental results demonstrate consistent improvements over strong baselines.

**Strengths:**

1. The overall narrative is coherent, with well-motivated objectives and a logically developed framework.
2. The method is clearly specified with well-presented definitions and equations, making the decomposition–fusion process straightforward and understandable.

**Weaknesses:**

1.  Figures 2 and 4 present case studies only in Chinese, which makes it difficult for readers unfamiliar with the language to understand the examples. Using machine translation may partially affect or obscure the linguistic nuances that illustrate the strengths of the proposed method.

2. The metrics partially rely on a hybrid LLM–human scoring, and the lack of detailed information, such as the specific prompts used, may make the results somewhat less persuasive and harder to replicate.


3. The discussion of the gating network’s computational cost is insufficient to demonstrate an efficiency advantage, and the presentation lacks intuitive visualization, such as a figure or table.

4.  Including pseudocode would help readers better understand the proposed method and clarify its implementation details, thereby improving reproducibility.

5.  Some presentation aspects could be improved. For example, using vector graphics (e.g., SVG) instead of bitmap formats (e.g., PNG) can prevent blurriness, particularly in Figures 2 and 4. Additionally, there is inconsistency in abbreviations: the full term "Retrieval-Augmented Generation" should appear only once at first use, with "RAG" used consistently thereafter. Currently, it is first written as "retrieval-augmented generation" and later as "Retrieval-Augmented Generation"; this should be corrected for clarity and coherence.

**Questions:**

1. The current comparison includes too few datasets and baselines, and the dataset is private, limiting reproducibility and external validation. It is worth considering using datasets [1][2], which allow evaluation without relying on LLMs by directly computing ROUGE-L and METEOR, and comparing with the methods in [3][4]. If this approach is not feasible, an explanation of the reasons would be appreciated.
2. Since efficiency is discussed, it would be valuable to clarify whether the approach can be accelerated using widely adopted tools like vLLM. This aspect is important for practical deployment and real-world system efficiency.
3. The absence of ablation experiments and hyperparameter analyses is notable. This omission makes it difficult to understand how different components or settings impact the model's performance. It would be beneficial to provide an explanation for why these studies were not included, as their absence can be somewhat confusing.

- [1]	Salemi A, Mysore S, Bendersky M, et al. Lamp: When large language models meet personalization[J]. arXiv preprint arXiv:2304.11406, 2023.
- [2]	Kumar I, Viswanathan S, Yerra S, et al. Longlamp: A benchmark for personalized long-form text generation[J]. arXiv preprint arXiv:2407.11016, 2024.
- [3]	Tan Z, Liu Z, Jiang M. Personalized Pieces: Efficient Personalized Large Language Models through Collaborative Efforts[C]//Proceedings of the 2024 Conference on Empirical Methods in Natural Language Processing. 2024: 6459-6475.
- [4]	Zhang J, Liu Y, Wang W, et al. Personalized Text Generation with Contrastive Activation Steering[J]. CoRR, 2025.
- [5]	Kwon W, Li Z, Zhuang S, et al. Efficient memory management for large language model serving with pagedattention[C]//Proceedings of the 29th symposium on operating systems principles. 2023: 611-626.

---

### Official Review · Reviewer_nGbb · 2025-11-01

**Soundness:** 2
**Presentation:** 3
**Contribution:** 2
**Rating:** 4
**Confidence:** 3

**Summary:**

The paper introduces an adaptive low-rank gated framework for large-scale personalized language style modeling. It decomposes author styles into twelve interpretable linguistic dimensions, such as semantics, syntax, and formality, and trains a separate module for each dimension. A dynamic gating network then fuses these modules with context-dependent weights to generate responses that reflect individual author styles. This design enables a single model to represent unique styles with zero-shot generalization, parameter efficiency, and interpretability. Experiments on a dataset show that their work outperforms baselines in style fidelity, accuracy, and adaptability to unseen styles.

**Strengths:**

The paper presents a scalable multi-style language modeling, a decomposition-fusion framework that adaptively combines multiple low-rank modules through a gated network. The idea of decomposing author style into interpretable, orthogonal linguistic dimensions (semantics, syntax, grammar, etc.) (not justified properly though) and learning specialized LoRA modules per dimension is practically significant. The method shows empirical results on a real-world dataset, outperforming baselines while remaining efficient and interpretable. The paper's contribution to parameter-efficient, interpretable personalization for large language models is promising.

**Weaknesses:**

1. The choice of exactly twelve “orthogonal” stylistic dimensions is arbitrary and unsupported by any established theories, or principled selection procedure or ablation demonstrating that twelve is optimal.
2. The empirical evaluation is not enough, lacks broader validation across diverse domains/languages.
2. Evaluation metrics rely partly on model-assisted scoring without sufficient human validation details though not-so specified expert annotation details are mentioned, but still evaluator bias and/or circularity assumptions should have been discussed in the paper.
3. The interpretability results are mostly qualitative and could be further strengthened by quantitative analyses linking gating weights to specific linguistic phenomena.
4. Ablation studies on the number of dimensions or gating design are not sufficient at all for discarding the uncertainity
5. The orthogonality constraint on learned global style embeddings may force irrelevant disentanglement that harms real-world stylistic nuance - that pov should have experimented or shown in the paper to discard any possibilities.
6. Reliance on synthetic style rewriting functions for LoRA pretraining risks teaching the adapters artifacts of the rewrites rather than genuine authorial style - no sort of negatation experiments are also not available.
7. The gating network’s design is under-justified and no architectural search or sensitivity analysis is provided to show it isn’t a brittle hyperparameter choice.
8. The paper reports marginal numeric gains (e.g., 4.28 vs 4.25) without statistical significance testing, leaving it unclear whether improvements are really meaningful.
9. KL regularization on LoRA is proposed but no ablation quantifies its trade-off between adaptability and catastrophic drift.
10. The time-cost comparison lacks resource-normalized profiling (memory, GPU utilization, batch throughput) to support efficiency claims.
11. Related works are cited but critical baselines like training-time multi-task adapters or prompt-tuning hybrids are not thoroughly compared.

**Questions:**

1. How justified is the twelve-dimensional decomposition, would different linguistic factor choices (e.g., fewer or non-orthogonal dimensions) yield similar performance?
2. Can the authors provide quantitative interpretability evidence showing correlation between each style dimension and measurable linguistic features in outputs?
3. How does LoRAMax perform when the base model is smaller (e.g., ≤7B parameters)? Does the gating mechanism remain effective under lower-capacity models?
4. What is the computational overhead of maintaining twelve LoRA modules compared to single- or cluster-based LoRA fusion in large-scale inference? This is important to understand the usability of the work.

---

### Note · Program_Chairs · 2026-01-17
**Submission Desk Rejected by Program Chairs**

The following references in this submission do not refer to real documents and/or have major errors in bibliographic information:

 Li Zheng et al. Personalizing dialogue generation with fine-grained style control. In EMNLP, 2021. URL https://aclanthology.org/2021.emnlp-main.517.pdf.